# Integrins regulate epithelial cell shape by controlling the architecture and mechanical properties of basal actomyosin networks

Carmen Santa-Cruz Mateos[1], Andrea Valencia-Expósito[1], Isabel M. Palacios[2], María D. Martín-Bermudo[1]*

1 Centro Andaluz de Biología del Desarrollo, Universidad Pablo de Olavide/CSIC/JA, Carretera de Utrera, Sevilla, Spain, 2 School of Biological and Chemical Sciences, Queen Mary University of London, London, United Kingdom

* mdmarber@upo.es

**Data Availability Statement:** All relevant data are within the manuscript and its Supporting Information files.

## Abstract

Forces generated by the actomyosin cytoskeleton are key contributors to many morphogenetic processes. The actomyosin cytoskeleton organises in different types of networks depending on intracellular signals and on cell-cell and cell-extracellular matrix (ECM) interactions. However, actomyosin networks are not static and transitions between them have been proposed to drive morphogenesis. Still, little is known about the mechanisms that regulate the dynamics of actomyosin networks during morphogenesis. This work uses the *Drosophila* follicular epithelium, real-time imaging, laser ablation and quantitative analysis to study the role of integrins on the regulation of basal actomyosin networks organisation and dynamics and the potential contribution of this role to cell shape. We find that elimination of integrins from follicle cells impairs F-actin recruitment to basal medial actomyosin stress fibers. The available F-actin redistributes to the so-called whip-like structures, present at tricellular junctions, and into a new type of actin-rich protrusions that emanate from the basal cortex and project towards the medial region. These F-actin protrusions are dynamic and changes in total protrusion area correlate with periodic cycles of basal myosin accumulation and constriction pulses of the cell membrane. Finally, we find that follicle cells lacking integrin function show increased membrane tension and reduced basal surface. Furthermore, the actin-rich protrusions are responsible for these phenotypes as their elimination in integrin mutant follicle cells rescues both tension and basal surface defects. We thus propose that the role of integrins as regulators of stress fibers plays a key role on controlling epithelial cell shape, as integrin disruption promotes reorganisation into other types of actomyosin networks, in a manner that interferes with proper expansion of epithelial basal surfaces.

## Author summary

Morphogenesis involves global changes in tissue architecture driven by cell shape changes. Mechanical forces generated by actomyosin networks and force transmission through adhesive complexes power these changes. The actomyosin cytoskeleton organises in

**Funding:** Research in our laboratories is funded by the Spanish Ministerio de Economía y Competitividad and the FEDER programme (BFU2013-48988-C2-1-P and BFU2016-80797-R) to M.D. M-B.) and by the Junta de Andalucía (Proyecto de Excelencia P09-CVI-5058). C. S-C M and A. V-E were supported by FPU and FPI Fellowships, respectively (Ministerio Español de Economía y Competitividad). IMP was supported by the BBSRC, the University of Cambridge and Queen Mary University of London. The funders had no role in study design, data collection and analysis, decision to publish, or preparation of the manuscript.

**Competing interests:** The authors have declared that no competing interests exist.

different types of networks, which localise to precise regions and perform distinct roles. However, they are rarely independent and, often, reorganisation of a given structure can promote the formation of another, conversions proposed to underlie many morphogenetic processes. Nonetheless, the mechanisms controlling actomyosin network dynamics during morphogenesis remain poorly characterised. Here, using the *Drosophila* follicular epithelium, we show that cell-ECM interactions mediated by integrins are required for the correct distribution of actin in the different actin networks. Elimination of integrins results in redistribution of actin from stress fibers into a new type of protrusions that dynamically emanate from the cortex and extend into the stress fibers. Changes in area protrusions correlate with bursts of myosin accumulated in stress fibers and constriction pulses of the cell membrane. We also found that integrin mutant cells show increased membrane tension and reduced basal cell surface. As these defects are rescued by eliminating the F-actin protrusions, we believe these structures prevent proper basal surface growth. Thus, we propose that integrin function as regulators of stress fibers assembly and maintenance controls epithelial cell shape, as its disruption promotes reorganisation into other actomyosin networks, conversions that interfere with proper epithelial basal surface expansion.

## Introduction

Forces generated by F-actin networks are important contributors to the generation of cell and tissue shape. The architecture and mechanical properties of the F-actin network are modulated by myosin II (MyoII) motors and actin binding proteins (reviewed in [1]). The molecular composition of contractile actin networks and bundles is highly conserved among eukaryotic species [2]. Nevertheless, their organisation and dynamics change across different cell types, their position within the cell and the differentiation state of the cell.

There are two main ways in which actomyosin networks can be organised within the cell, as a cortical meshwork below the plasma membrane or as stress fibers. Studies over the last decade have assigned distinct roles for these two types of networks. While pulsatile contraction of cortical actomyosin networks has been mainly implicated in the cell shape changes underlying key morphogenetic processes, such as gastrulation and neural tube formation (reviewed in [3]), stress fibers have been largely involved in cell adhesion, migration and mechanosensing [4]. During morphogenesis, cells change the way they organise their actin networks in response to intracellular signals. For example, a change in actin organisation from a cortical rearrangement into stress fibers is observed when cells exit mitosis or naïve pluripotency, or during epithelia to mesenchyme transitions (reviewed in [5]). In addition, transitions between networks also depend on the way cells interact with each other and with the extracellular environment. Thus, while cell-cell interactions promote cortical actin organisation, cell-ECM adhesion stimulates stress fibers formation. During morphogenesis, conversions between these two different actin networks need to be finely regulated in space and time, as misplaced or untimely transitions could affect the proper formation of organs and tissues. Still, little is known about the mechanisms that guarantee controlled transitions during morphogenesis.

The follicular epithelium (FE) of the adult *Drosophila* ovary provides an excellent model system to study the contribution of cell-ECM interactions to the organisation of actin networks during morphogenesis. The *Drosophila* ovary is composed of 16–18 structures called ovarioles [6]. Each ovariole contains a germarium at their anterior end and progressively older egg chambers towards the posterior end. Each egg chamber is composed of a cyst of 15 nurse

cells and one oocyte enveloped by a single layer of follicle cells (FCs), which constitutes the FE [7]. Newly formed egg chambers are round and go through 14 developmental stages (from S1 to S14) to eventually give rise to mature eggs [7]. At the time that the egg chamber buds off from the germarium, approximately 80 FCs enclose the germline cyst [7]. FCs continue to divide mitotically until S6 when they exit the mitotic cycle and switch to an endocycle [8]. Between S7-10, FCs undergo three rounds of endoreplication, become polyploid and increase their size. The apical side of FCs faces the germline, while their basal surface contacts a specialised ECM called basement membrane, which encapsulates the egg chamber [9]. Throughout oogenesis, F-actin organises at the basal side of FCs in three different types of networks, a cortical meshwork, planar-polarised protrusions and polarised stress fibers (Fig 1A; [10–12]). Planar polarised protrusions and polarised stress fibers show dynamic behaviours throughout oogenesis. From early stages of oogenesis until the end of S8, the FE rotates around its anterior-posterior axis, a process termed "global tissue rotation" [13, 14]. During this process, two types of planar polarised protrusions have been identified, one typical of migrating cells containing lamellipodia and filopodia [10], and a second one termed whip-like structure [11]. Both protrude from the cell membrane but differ in their dynamics and localisation within the cell. Filopodia and lamellipodia localise to the leading edge of migrating FCs and show protrusive activity, extending in the direction of the movement. In contrast, whip-like structures are restricted to tricellular junctions and show flagella-like dynamics, propelling contrary to the direction of FE rotation. In addition, filopodia and lamellipodia are shorter than whip-like structures. The mechanisms regulating the assembly and maintenance of these different types of networks and the relationship among them throughout oogenesis remain poorly understood.

Cell culture studies have revealed a key role for integrins in the production and organisation of stress fibers, especially during cell migration [15]. Integrins are heterodimeric receptors composed of an α and a β subunit. While in vertebrates there are at least 8 β subunits and 18 α subunits, in *Drosophila* there are only two β subunits, βPS and βν, and five α subunits, αPS1 to αPS5. The βPS subunit, encoded by the gene *myospheroid* (*mys*), is the only β chain present in the ovary and localises at the ends of actin stress fibers and whip-like structures (Ng et al., 2016;[12, 16, 17]). The role of integrins on the organisation of actomyosin fibers on the basal side of FCs remains a bit controversial. On one hand, Delon and Brown found that clones of FCs lacking the βPS subunit showed increased basal F-actin bundles when compared to adjacent wild type cells [12]. This led them to propose that, in this context, integrins were not required to generate stress fibers but to reduce their number [12]. This differs from the proposed role for integrins in the formation of stress fibers in cultured cells [15]. In contrast, and in agreement with results from cell culture experiments, studies aimed at understanding the regulation of basal MyoII oscillations showed that reducing the levels of integrins, either by RNAi, expression of β-integrin mutant forms or optogenetics, resulted in a reduction in basal F-actin and MyoII intensities and oscillation periods during S9 to S10B [18, 19]. Finally, an analysis of the role of integrins on the formation and maintenance of the other types of F-actin networks present in FCs remains missing.

Here, we have used real-time imaging, laser ablation and quantitative image analysis to show that integrins are required to maintain the architecture and mechanical properties of basal actomyosin networks. This, in turn, is essential to regulate cell shape. Loss of integrins in FCs leads to: 1) an increase in the number of whip-like structures, 2) a reduction in stress fibers and 3) a reorganisation of F-actin into a new type of membrane protrusions, which emanate from the basal cortex and extend into the cell center, overlying the medial basal actomyosin fibers. Furthermore, these new F-actin protrusions are dynamic and changes in protrusion area correlate with both changes in basal myosin levels and constriction pulses of the cell

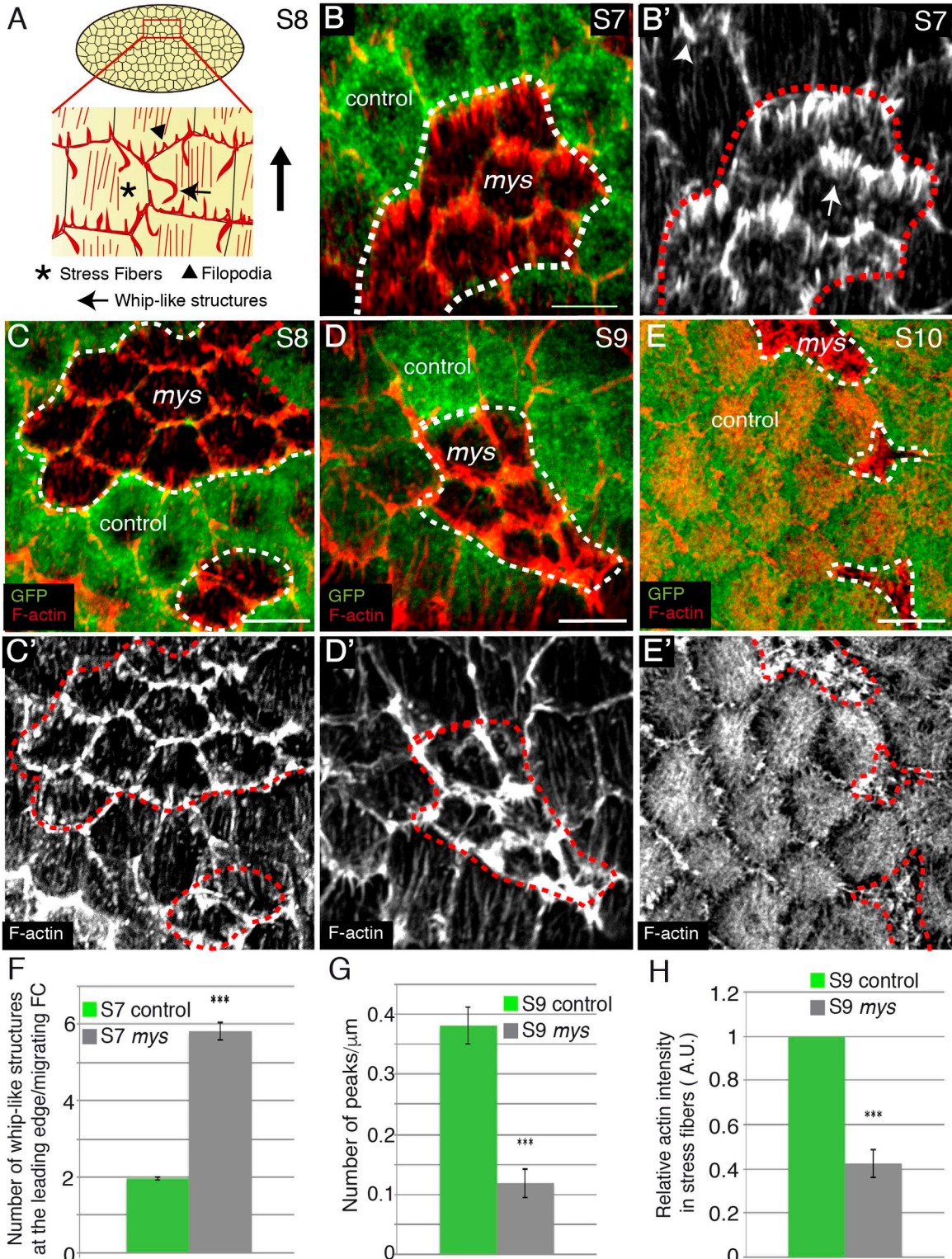

**Fig 1. Integrins regulate whip-like structures and stress fibers formation. (A)** Schematic drawing of a S8 egg chamber illustrating the different types of actin organisations found on the basal side of FCs. **(B-E')** Basal surface view of mosaic S7 (B, B'), S8 (C, C'), S9 (D, D') and S10 (E, E') egg chambers containing *mys* FC clones, stained for anti-GFP (green) and Rhodamine Phalloidin to detect F-actin (red). **(B, B')** *mys* FCs (GFP-negative) contain more whip-like structures (arrow in B') than control FCs (GFP-positive, arrowhead in B'). **(C-E')** Stress fiber number diminishes progressively from S8-10 in *mys* FCs. **(F)** Quantification of the number of whip-like structures at

the leading edge of S7 control and *mys* migrating FCs. (**G**) Quantification of the number of actin fibers per μm in S9 control and *mys* FCs. (**H**) Quantification of relative F-actin intensity in stress fibers in S9 control and *mys* FCs. The statistical significance of differences was assessed with a t-test, *** P value < 0.0001. All error bars indicate s. e. Scale bars, 5 μm. The dotted white and red circles indicate area occupied by clones of mutant cells.

membrane. Finally, we found that integrin mutant FCs show reduced basal surface and increased membrane tension, two traits that can be rescued by blocking the formation of the membrane actin-rich protrusions. Altogether, we propose that integrin function as regulators of stress fibers assembly and maintenance controls epithelial cell shape, as its disruption promotes reorganisation into other types of actomyosin networks, conversions that interfere with proper expansion of epithelial basal surfaces.

## Results

### Integrins regulate the formation and dynamics of basal actin networks

To deepen in our understanding of the role of integrins in the formation and dynamics of basal actin networks, we generated mosaic egg chambers containing clones of FCs homozygous for the null allele $mys^{XG43}$. Cell culture experiments have shown that loss of contact with the ECM mediated by integrins results in programmed cell death. However, we found that elimination of integrins in main body FCs did not induce cell death, as tested using an antibody to cleaved Dcp-1 (ec = 14, S1 Fig). To visualise basal actin networks, we used the F-actin marker Rhodamine Phalloidin. We found that $mys^{XG43}$ mutant FCs (from now on *mys* FCs) showed increased numbers of basal actin-rich protrusions resembling whip-like structures compared to controls (FCs analysed for each cell type, n = 24; egg chambers analysed, ec = 8, Fig 1B and 1B'). To test whether they were in fact whip-like structures and to quantify them unambiguously, we performed live imaging of mosaic S7 egg chambers expressing either a LifeActin-YFP under a ubiquitous promoter, Ubi-LifeActinYFP, generated in this study (see Materials and Methods, S1 Movie), or the myosin regulatory light chain Spaghetti-Squash tagged with GFP (Sqh-GFP, S2 Movie, [20]). We found that the ectopic basal actin-rich protrusions observed in *mys* FCs were indeed whip-like structures, as they moved like flagella against the direction of rotation (S1 Movie) and did not contain myosin (n = 50, ec = 8, S2 Fig, S2 Movie). Quantification analysis of the *in vivo* images showed that while control FCs contained two whip-like structures at the leading edge (Fig 1F, n = 24, ec = 8), one at each tricellular junction, *mys* FCs contained on average more than 5 (Fig 1F, n = 24, ec = 8). Thus, we propose that integrins are required to restrain the number of whip-like structures.

Next, we studied the function of integrins in the formation and maintenance of stress fibers. It has been previously reported that downregulation of β integrin function, by either RNAi or optogenetics, reduces the intensity and oscillation period of basal F-actin and myosin signals during S9 and S10 [18, 19]. To further characterise integrin function in stress fiber morphogenesis in FCs, we analysed the density and morphology of the actomyosin fibers in control and *mys* FCs throughout oogenesis. We found that already at S8 the density of actomyosin fibers was reduced in *mys* FCs compared to controls (n = 36, ec = 8, Fig 1C and 1C', S3A and S3A' Fig). This phenotype worsened as oogenesis progressed (n = 25, ec = 8, Fig 1D and 1E', S3B–S3C' Fig), so that by S10 basal stress fibers were hardly detectable (n = 32, ec = 8, Fig 1E and E' and S3C and S3C' Fig). We quantified this phenotype by measuring the number and morphology of stress fibers at S9, when they were fully extended in control FCs and still visible in the *mys* FCs. We measured F-actin staining intensity across a bar centred at the basal side of FCs and identified peaks in which the fluorescence intensity exceeded one standard deviation below the mean intensity in the control (see Materials and Methods). We found that the

number of peaks per micrometre was lower in *mys* FCs compared to controls (n = 25, ec = 8, Fig 1G). Likewise, the overall intensity of F-actin in stress fibers in the mutant FCs was reduced by 60% with respect to controls (n = 23, ec = 8, Fig 1H). Furthermore, we found that myosin levels were also reduced by 40% in mutant FCs compared to controls (n = 27, ec = 8, S3D Fig). Finally, using live imaging of mosaic S10 egg chambers expressing the membrane marker Resille-GFP [21] and either our Ubi-LifeActinYFP (S4 Fig, S3 Movie) or Sqh-mCherry (S5 Fig, S4 Movie), we found that the preferential pulsation periods of both F-actin (n = 10, ec = 8) and myosin (n = 9, ec = 8) were reduced in mutant cells compared to controls (n = 8, ec = 8, S4C–S4E Fig and S5C–S5E Fig), in agreement with a previous report [19]. In addition, we found that mutant cells displayed a more stochastic behaviour than controls (S4C–S5E Fig and S5C–S5E Fig). Altogether, our results show that integrins regulate the formation, maintenance and dynamics of basal actomyosin networks from early stages of oogenesis.

## Elimination of integrins results in F-actin reorganisation

Adherent mouse embryonic fibroblasts that gradually detach from the substrate redistribute their F-actin from stress fibers to a more cortical position. Because integrins mediate fibroblast adhesion to the substrate, this result points to a role of integrins in the organisation of F-actin [22]. However, a previous report indicated that the decrease in F-actin levels found in the stress fibers of integrin mutant FCs was not accompanied by a clear increase in the cortex, a result that led the authors to suggest that cell-matrix adhesion might control F-actin intensity, but not its distribution in FCs [19]. In contrast and similar to the case of fibroblasts in culture, we found that S9 integrin mutant FCs (n = 22, ec = 5) showed a strong accumulation of F-actin near the cortex (S6A and S6A' Fig). Quantification of F-actin intensity along cell-cell contacts revealed a two-fold increase in cortical F-actin in *mys* FCs compared to controls (n = 22, ec = 5, S6C and S6D Fig, see Materials and Methods). We also observed that all mutant cells within the clone, regardless of whether they were surrounded by either control or mutant cells, displayed increased levels of F-actin at basal cell edges, suggesting that this phenotype was cell autonomous (S6A and S6A' Fig). This was specific to the basal side, as no difference was found apically (n = 22, ec = 5, S6B and S6B' Fig). These results lead us to suggest that, in addition to its role in stress fiber formation, integrins can regulate F-actin redistribution in FCs.

We then characterised the organisation and behaviour of the dense basal junctional F-actin found in integrin mutant FCs. To do this, we performed live imaging of mosaic S10 egg chambers expressing the membrane marker Resille-GFP [21] and either our Ubi-LifeActinYFP or Sqh-mCherry. Time-lapse imaging resolved the dense cortical build-up of F-actin observed in *mys* FCs into a new type of actin-rich membrane protrusions that: 1) emerged from the cell membrane; 2) projected towards the cell center, overlying the medial basal actomyosin fibers (S5 and S6 Movies); and 3) did not contain myosin (S6 Movie, Fig 2A–2D). These actin rich protrusions differed from both, whip-like structures, which show flagella dynamics but not protrusive activity [11], and from the filopodia associated to egg chamber rotation, which are restricted to the leading edge and reach out over the basal surface of an adjacent cell [10]. Furthermore, the simultaneous quantification of myosin level oscillations and total area occupied by actin-rich protrusions over time showed that myosin accumulation correlated with increased protrusion area in mutant FCs (n = 50, ec = 9, Fig 2B–2F, S6 Movie). In addition, and similar to control FCs, in which the rate of myosin accumulation correlates with the rate of basal surface contraction [18], we found that higher myosin levels corresponded with higher contraction of the basal surface in integrin mutant FCs (n = 50, ec = 9, Fig 2F, S6 Movie). However, while the variation of the basal surface area of wild type FCs was shown to be highly polarised, being five-time higher in the D-V axis than in the A-P axis [18], we observed that

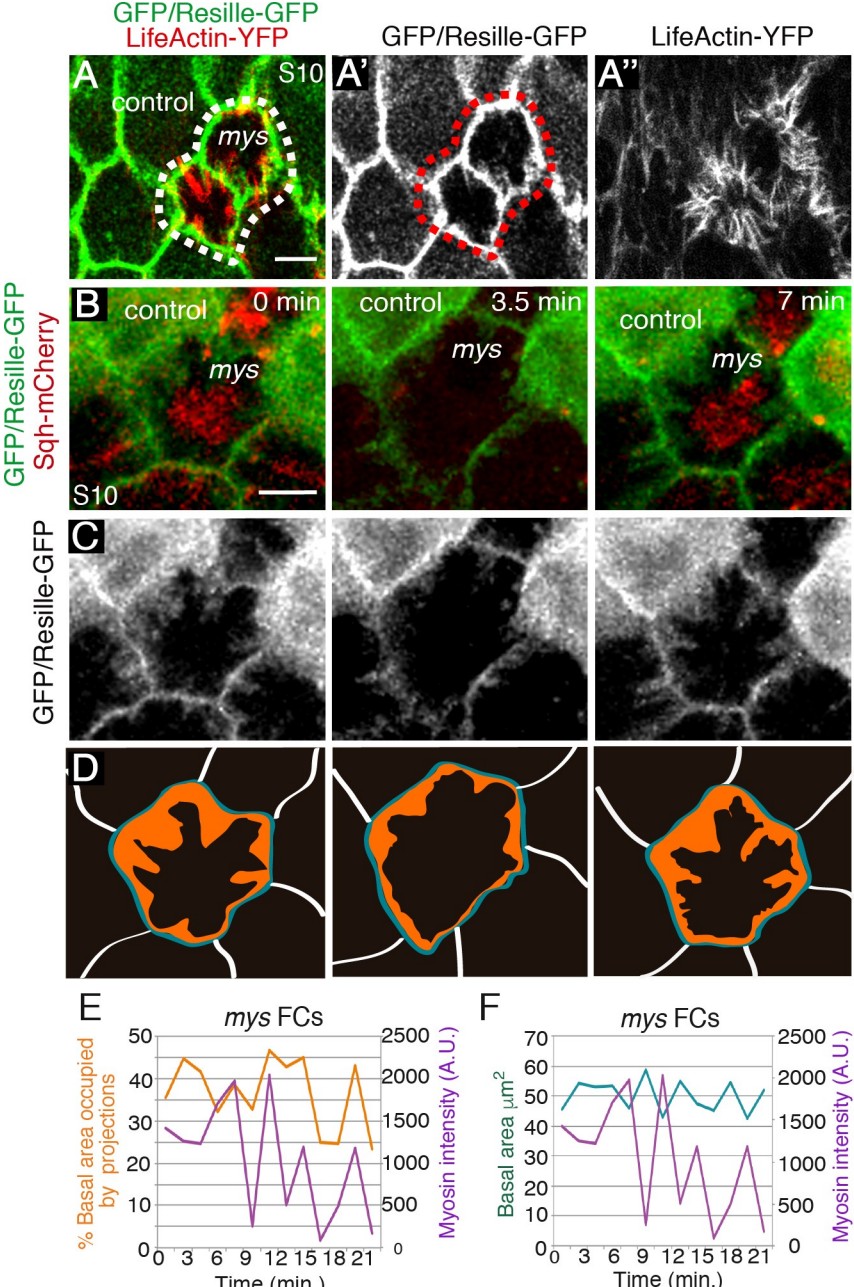

**Fig 2. Loss of integrins results in reorganisation of the basal actin cytoskeleton. (A-A")** Confocal micrographs of *mys* FC clones in living egg chambers expressing LifeActin-YFP (red) and the membrane marker Resille-GFP (green). *mys* FCs (GFP-negative) form abnormal actin-rich protrusions. **(B-D)** Time-lapse series of one representative *mys* FC (GFP-negative) labelled with Sqh-mCherry (red) and Resille-GFP (green). **(D)** The total area occupied by projections at different time points is coloured in orange. In blue, outline of the basal surface of the cell. **(E, F)** Simultaneous quantification of basal myosin changes and % of total basal surface occupied by projections **(E)** or total basal area **(F)** in one representative *mys* FC. Scale bars, 5 μm.

the reduction in basal surface area in mutant cells was almost symmetrical, with a ratio of D-V/A-P length change over time ~1.2 (n = 50, ec = 9). Taken together, the correlation between basal myosin accumulation, increased protrusion area and basal surface reduction

observed in *mys* FCs supported the notion that the three phenomena might be related. Furthermore, it also suggests that the symmetrical basal surface contractions found in mutant FCs could be due to the activity of the ectopic protrusions, which in turn could be regulated by the medial actomyosin fibers. A direct outcome of this hypothesis is that membrane tension between two adjacent mutant FCs should be higher than the tension between two wild type cells or between a wild type cell and a mutant one, as protrusions from two mutant adjacent cells would be pulling in opposite directions. This led us to investigate whether elimination of integrins affected membrane tension.

### Integrin mutant FCs show increased membrane tension

Laser ablation of cell-cell boundaries is an effective tool to measure tension at junctional membranes [23]. Thus, we performed laser ablation experiments using a UV laser beam to sever plasma membranes and the cortical cytoskeleton on the basal side of either S9 control or *mys* FCs (see Materials and Methods). The behaviour of cell membranes, visualised with Resille-GFP [21], was monitored up to fifteen seconds after ablation. As a consequence of the cut, cortical tension relaxed and the distance between the cell vertices at both sides of the cut increased. Because the velocity of retraction is affected by cytoplasmic viscosity [24], we assumed viscosity in *mys* and control FCs to be the same. To minimise potential effects due to an anisotropic distribution of forces in the FE, cuts were all made perpendicular to the AP axis and in the central region of fourteen independently cultured egg chambers (Fig 3A and 3B, S7 and S8 Movies). We found that the initial velocity of vertex displacement between two adjacent mutant FCs (1.26μm/sec, n = 14, ec = 14) was two times higher than the velocity found between two control cells (0.58μm/sec, n = 14, ec = 14, Fig 3C, S7 and S8 Movies). In addition, vertex displacement over time was also significantly greater in mutant cells compared to controls (n = 14, ec = 14, Fig 3D, S7 and S8 Movies). Finally, tension at boundaries between mutant and control cells (0.61μm/sec, n = 24, ec = 24) was similar to the tension observed between two control cells (Fig 3C and 3D). These results allowed us to conclude that membrane tension increases at the boundary between two adjacent integrin mutant FCs.

### Morphological consequences of integrin elimination in FCs: defective basal surface expansion

As changes in membrane tension are known to regulate cell shape [25], we next tested whether integrins were required to regulate cell shape in FCs. Using Resille-GFP to outline individual cells, we found that the basal surface of S10 *mys* FCs (n = 63, ec = 10) was smaller than that of controls (n = 63, ec = 10, Fig 4A, 4C and 4D), whereas we observed no significant change either on the apical side or the height (n = 70, ec = 10, Fig 4B, 4C, 4E and 4F).

When cultured cells detach from the ECM, tensile loads on the cytoskeleton become unbalanced, stress fibers contract and cells shrink and adopt a rounded morphology [26]. In this context, the reduced basal surface found in S10A *mys* FCs could be due to cell shrinkage. Alternatively, as basal FCs surface expand during S7-10 [7, 27], the diminished basal surface observed in mutant FCs could also be due to defective surface expansion. To distinguish between these two possibilities, we measured surface area of control and mutant FCs throughout oogenesis. We found that the basal surface of mutant cells (n = 46, ec = 5) did not increase from S6 to S10 at the same rate as that of controls (n = 46, ec = 5, Fig 4G). In fact, control cells grew their basal surface 1.7 and 1.8 times from S8-9 and from S9-10, respectively, while mutant cells grew their basal surfaces 1.5 and 1.3 times from S8-9 and from S9-10, respectively (Fig 4G). To test that the failure of *mys* FCs to increase their size from S6-10 was not due to inappropriate endoreplication, we measured the nuclear size of control and mutant FCs. We found

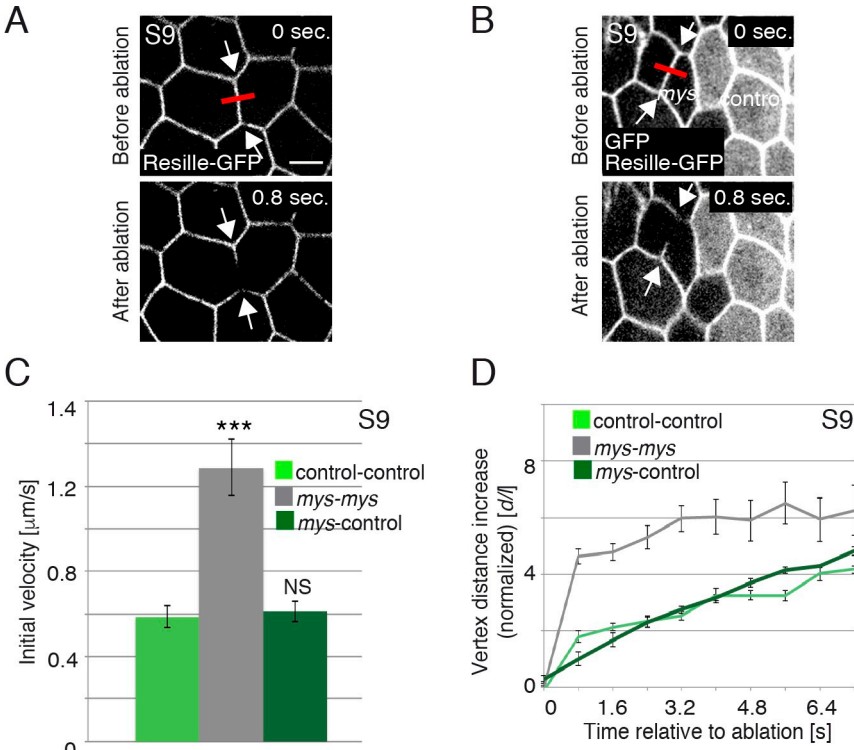

**Fig 3. Loss of integrins in FCs results in increased membrane tension. (A, B)** Images of life S9 wild type **(A)** and mosaic egg chambers containing *mys* FC clones (GFP-negative) **(B)**, expressing Resille-GFP, before and after single-cell bonds are ablated. Red bar and arrows indicate the point of ablation and the vertexes displaced, respectively. **(C)** Quantification of initial velocity of vertex displacement and **(D)** vertex displacement over time of the indicated ablated bonds. The statistical significance of differences was assessed with a t-test, *** P value < 0.0001. All error bars indicate s. e. Scale bars, 5 μm.

that the nuclear size of *mys* FCs (n = 37, ec = 10) was similar to that of controls (n = 37, ec = 10, Fig 4H), in agreement with previous findings showing a normal BrdU incorporation pattern in *mys* FCs in contact with the germline [28]. Altogether, these results lead us to propose that the reduced basal surface observed in the mutant cells is not a failure in global growth, but a specific requirement of integrins for proper expansion of their basal surface.

## Integrins regulate basal surface area by controlling F-actin organisation

Stress fibers and cortical tension, both of which are affected in integrin mutant FCs, have been proposed to regulate cell shape (reviewed in [29]). To determine the cause of the defects in basal surface growth due to integrin elimination, we decided to block the formation of the actin-rich protrusions in *mys* FCs. Suppression of the Abelson interacting protein (Abi*)* in FCs leads to a complete elimination of filopodial protrusions and whip-like structures and a strong disorganisation of stress fibers [10, 11]. Consistently, we found that RNAi-depletion of *abi* in all FCs, using the *traffic jam*-Gal4 (*tj*-Gal4) driver (tj>*abi*RNAi; [30]), abolished the formation of all actin protrusions, including the basal actin-rich protrusions formed in *mys* FCs (n = 34, ec = 5, Fig 5A–5C). Interestingly, this experimental condition was able to rescue both the reduction in basal surface (n = 34, ec = 5, Fig 5A–5B', 5D and 5E) and the increase in membrane tension typical of *mys* FCs (Fig 5E). Thus, while the initial velocity of vertex displacement between two adjacent *mys* mutant FCs was around 1.28μm/sec (Fig 5E, n = 14, ec = 14), that found between two *mys*;*abi*RNAi cells was around 0.38μm/sec (Fig 5E, n = 15, ec = 15),

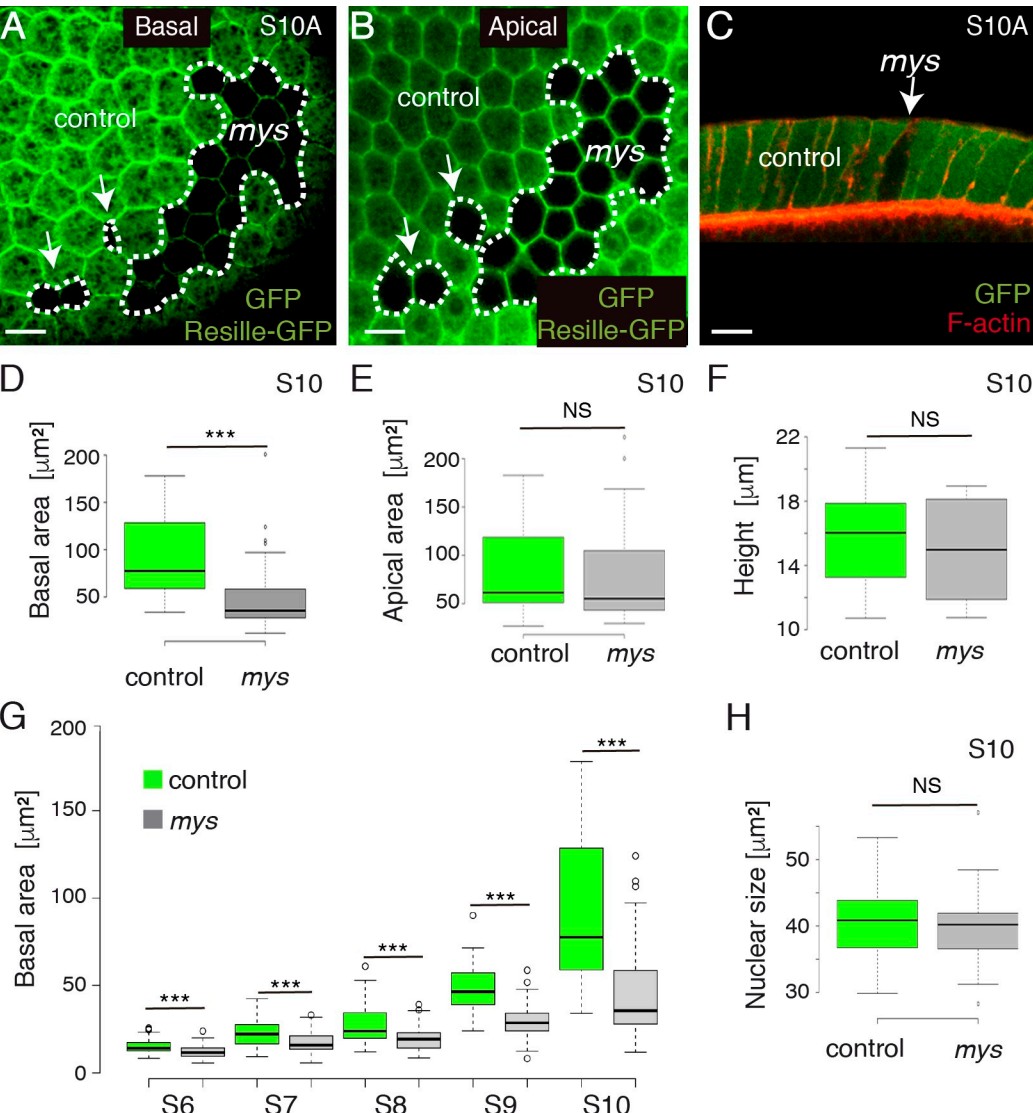

**Fig 4. *mys* mutant FCs show defective basal surface expansion.** Basal (**A**) and apical (**B**) surface views of a S10A mosaic egg chamber containing *mys* FC clones (GFP-negative) and expressing the cell membrane marker Resille-GFP, stained with anti-GFP. (**C**) Lateral view of a mosaic FE stained with anti-GFP (green) and Rodamine Phalloidin to detect F-actin (red). (**D, E, F**) Box plots of the basal surface (**D**), apical surface (**E**) and height (**F**) of control (green) and *mys* (grey) S10 FCs. (**G**) Box plot of the basal area of control (green) and *mys* (grey) FCs at different stages of oogenesis. (**H**). Box plot of the nuclear size of control (green) and *mys* (grey) S10 FCs. The statistical significance of differences was assessed with a t-test, *** P value < 0.0001. Scale bars, 10μm.

closer to that found between two adjacent wild type FCs (0.59μm/sec, Fig 5E, n = 16, ec = 16). These results strongly suggest that the defects in basal surface growth observed in *mys* FCs were not due to defective stress fibers, but to the re-organisation of F-actin into protrusions.

Based on the correlation between high levels of basal myosin and the symmetrical decrease in basal surface observed in *mys* FCs, we have proposed above that these two phenomena could be connected. To test this hypothesis, we interfered with myosin contractility in *mys* FCs by expressing a dominant negative form of the non-muscle myosin heavy chain *zipper* tagged with GFP (*zip^{DN}*-GFP, [31]). Cell culture studies have shown that myosin-derived tension can control actin filament assembly in migrating cells [32]. Similarly, expression of *zip^{DN}*-GFP in

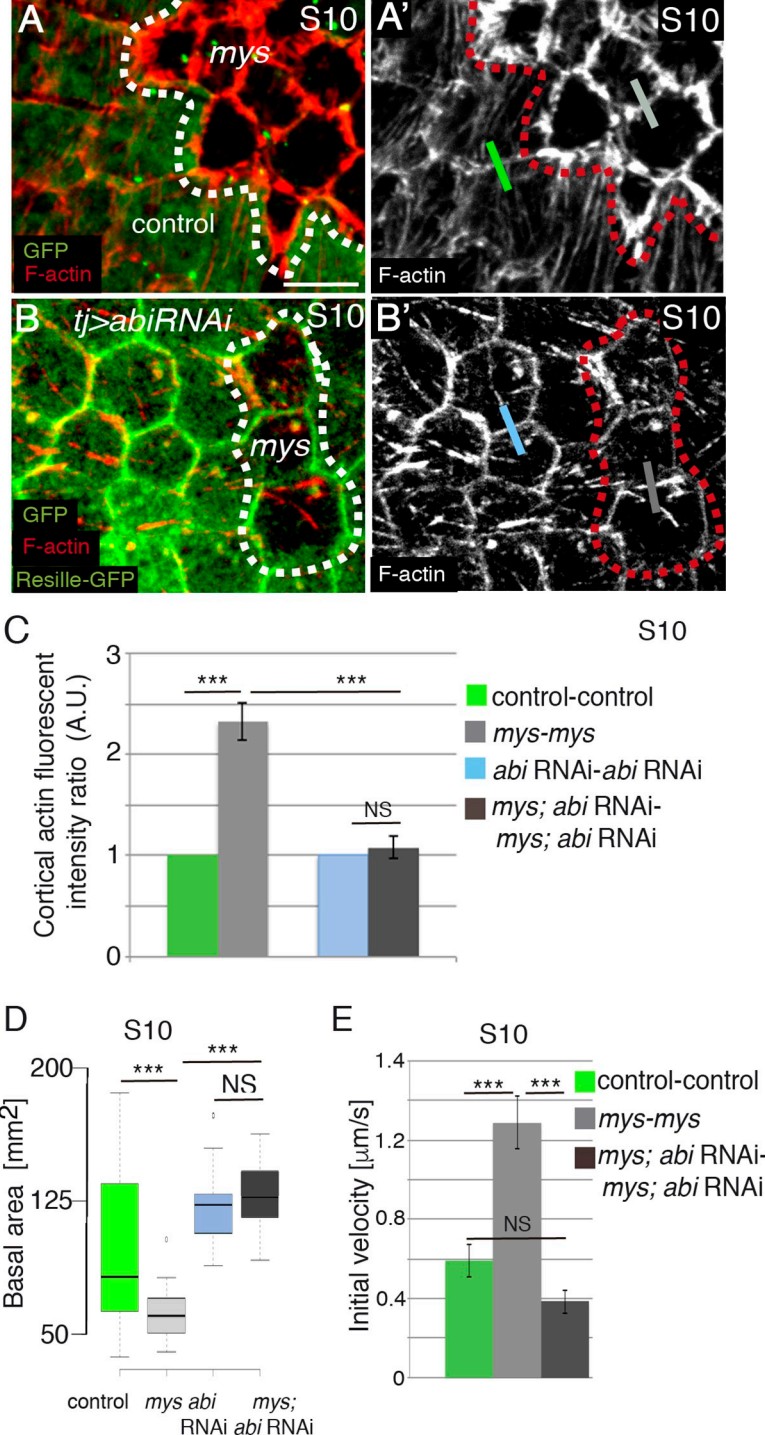

**Fig 5. Integrins regulate basal surface area by controlling F-actin levels at cell edges.** (**A, A', B, B'**) Basal surface view of S10 mosaic FE containing *mys* FC clones (GFP-negative) and (**B, B'**) expressing an *abi* RNAi (*tj>abiRNAi*), stained for anti-GFP (green) and Rhodamine Phalloidin to detect F-actin (red). (**C**) Quantification of relative F-actin intensities along boundaries between cells of the indicated genotypes. (**D**) Quantification of the basal area of S10 FCs of the designated genotypes. (**E**) Quantification of initial velocity of vertex displacement of the indicated ablated cell bonds. The statistical significance of differences was assessed with a t-test, *** P value < 0.0001. All error bars indicate s. e. Scale bars, 5 μm.

FCs decreased F-actin levels at stress fibers and at basal cell edges (n = 26, ec = 6, S7A and S7A' Fig). In addition, we found that expression of $zip^{DN}$-GFP in integrin mutant FCs (*mys*; *tj*> $zip^{DN}$-GFP) was able to rescue their reduced basal surface (n = 26, ec = 6, S7B and S7B' Fig). This result supports our idea that forces generated by myosin contractility could in principle contribute to the inability of integrin mutant FCs to expand their basal surface properly. However, as expression of $zip^{DN}$-GFP in FCs decreased F-actin levels in all actin structures, this experiment does not allow us to clarify the specific contribution of actomyosin contractility to the defects in basal surface growth observed in integrin mutant FCs.

### Elimination of integrins in groups of FCs affects cytoskeletal reorganisation in neighboring control cells

As reported above, the correct growth during mid-oogenesis of the basal surface of FCs requires integrin function. Our analysis of later egg chambers (stage 10B) showed that 70% of the mutant clones (n = 28, ec = 12) contained cells in which the basal area was hardly visible (Fig 6A–6B'). This extreme reduction in basal surface was better appreciated in cross sections of egg chambers stained with an antibody against the lateral marker Discs Large (Dlg) (Fig 6C and 6C'), or in 3D reconstructions of control and mutant FCs (Fig 6D). Thus, it seemed as if the reduction in the basal area characteristic of *mys* FCs progresses over time.

Basal stress fibers are randomly oriented at this point in control S10B FCs (yellow asterisks in Fig 6E', [12]). In contrast, we noticed that control S10B FCs surrounding *mys* FCs polarised their basal stress fibers towards the mutant cells. Quantification of this phenotype revealed that control cells re-orient their stress fibers around clones of mutant cells in 80% of the cases analysed (n = 15, ec = 9). Furthermore, this behaviour was observed in 100% of the cases if control cells surrounded mutant ones with extremely reduced basal surface, suggesting that stress fibers reorientation in control cells relates to reduced basal surface in mutant cells (Fig 6B and 6E). There are at least two alternative explanations for these results. First, stress fibers polarisation can be driven by an anisotropic cell spreading in response to external stimuli (as shown in cell culture experiments) [33]. In this context, wild type FCs surrounding mutant ones could sense free ECM space left by the mutant cells and respond by spreading and reorganising their actin cytoskeleton. Second, we have shown here that control and mutant cells have different cortical tension. This could generate a mechanical stress in surrounding wild type cells leading to their stretching and stress fiber reorganisation. To test the first possibility, we performed live imaging of mosaic S10B egg chambers expressing Resille-GFP and found that the basal surface of control FCs contacting mutant ones seemed to spread anisotropically over the mutant cells (n = 50, ec = 8, Fig 6F and 6F', S9 Movie). This was never observed in mosaic egg chambers containing control GFP clones (Fig 6F and 6F', S6 Movie). As for the second possibility, one would expect expression of *abi* RNAi, which blocks formation of protrusions in mutant cells (see above), to restore orientation of stress fibers in adjacent wild type cells. Unfortunately, this could not be consistently tested due to the strong disorganisation of basal actin bundles observed in *abi* RNAi FCs (Fig 5B', [11]). Thus, we propose that the reduction of the basal surface observed in late S10B mutant FCs could arise from the activation of spreading capacity and/or a mechanical response in surrounding wild type cells.

### Discussion

The actomyosin cytoskeleton organises in different types of networks within cells, including stress fibers or cortical arrangements. Each type of network localises to a precise region of the cell where it performs a distinct function. However, actin networks are highly dynamic and

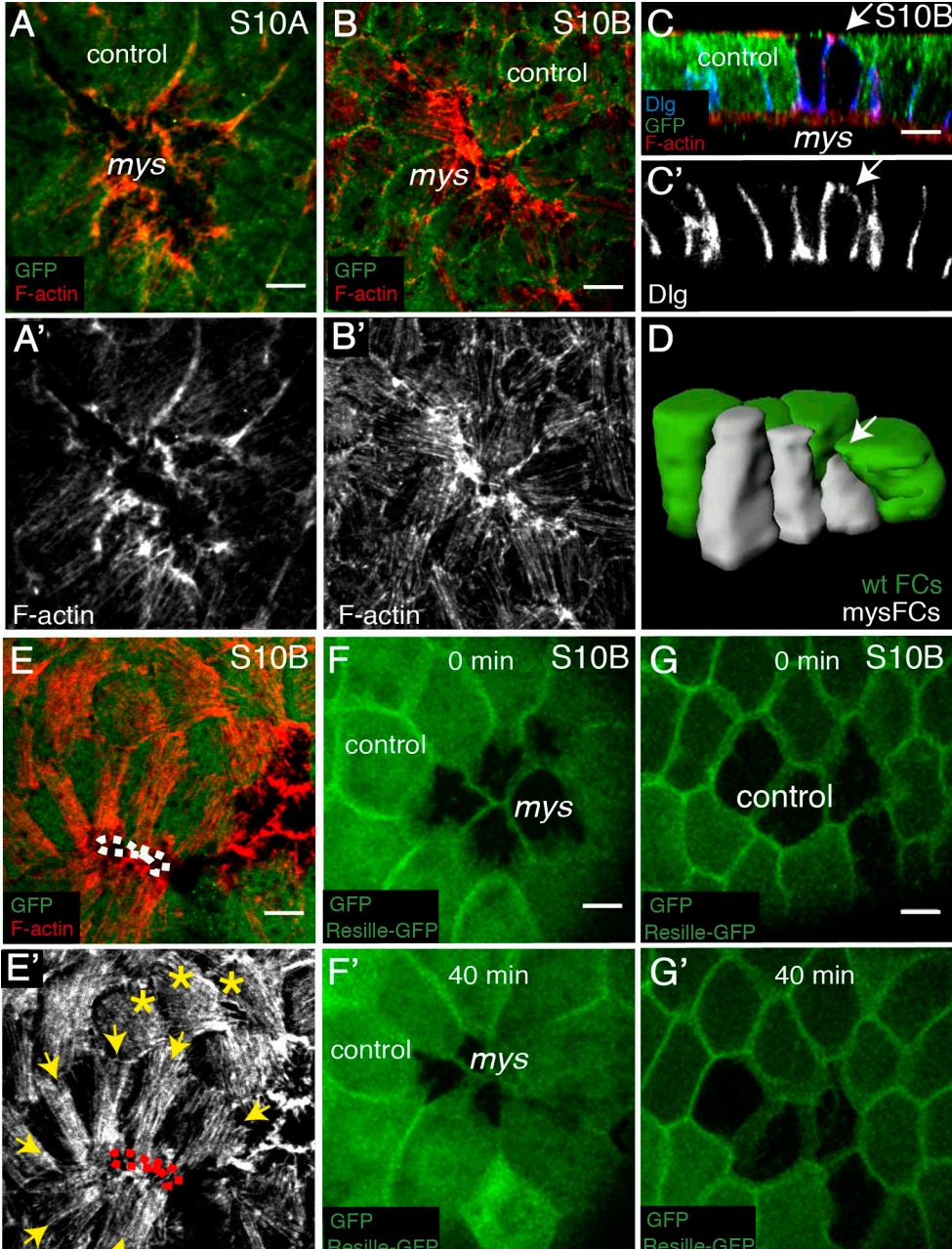

**Fig 6. Elimination of integrin in FCs disrupts cytoskeletal organisation in neighbouring control cells. (A, B)** Basal surface view of S10A **(A, A')** and S10B **(B, B')** mosaic follicular epithelia containing *mys* FC clones (GFP-negative), stained with anti-GFP (green) and Rhodamine Phalloidin to detect F-actin (red). **(C)** Lateral view of a S10 mosaic egg chamber stained with anti-GFP (green), Rhodamine Phalloidin (red) and anti-Dlg (basolateral polarity marker Discs large, blue). **(D)** 3D reconstruction of *mys* FCs and surrounding control cells. Arrows in **C** and **D** point to the basal surface of a mutant FC. **(E, E')** Basal surface view of a S10B mosaic FE containing *mys* FC clones (GFP-negative). Yellow arrows and asterisks mark control FCs contacting control and *mys* FCs, respectively. **(F-G')** Confocal images of live S10B mosaic egg chambers containing *mys* **(F)** or GFP **(G)** clones and expressing the cell membrane marker Resille-GFP. Images were taken with a 40 minutes interval. Scale bars, 5µm.

transitions between stress fiber and cortical organisations seem to drive key morphogenetic processes (reviewed in [5]). These transitions depend on intracellular signals and on cell-cell

and cell-ECM interactions. Here, we show that cell-ECM interactions mediated by integrins are required for the proper assembly and maintenance of stress fibers in FCs. Elimination of integrins from FCs leads to the reorganisation of F-actin from stress fibers into dynamic cortical protrusions, which we show interfere with proper expansion of the basal surface. Thus, our results show that equally important to trigger transitions between actin networks is to restrain them and suggest that integrins could act as supervisors of actomyosin network transitions during epithelia morphogenesis.

Most of our understating of the role of integrins on the assembly and dynamics of stress fibers comes from studies on cell migration. These studies have shown that even though the integrin-containing adhesion sites, focal adhesions, and stress fibers are two distinct structures with clear different functions, they are highly interdependent. During cell movement, recruitment of actin into stress fibers is impaired when focal adhesion proteins are eliminated. Similarly, focal adhesions rapidly disassemble when stress fibers are disrupted [34, 35]. However, little is known about the role of integrins in the assembly and dynamics of stress fibers during morphogenesis. Studies using the stress fibers on the basal side of *Drosophila* FCs have tried to address this issue [18, 19]. They have focused on the role of integrins on the dynamics of stress fibers, showing that integrins control basal myosin oscillations. Here, we find that elimination of integrins also causes a reduction in both myosin and F-actin levels in stress fibers before oscillations start. These results strongly suggest that, in static contractions, similar to what happens during cell migration, integrins play a key role in the nucleation and maintenance of actomyosin stress fibers. Cell culture experiments have also shown that stress fibers contract when separated from their focal adhesions [36]. Likewise, we find that, at the time stress fibers start to contract in FCs, they shorten and collapse in the absence of integrins. We believe this could be due to the fact that the balance in tension in the stress fibers -due to mechanical resistance of the focal adhesions to which they are attached to- is disrupted in integrin mutant cells. We also find that the detached actomyosin fibers found in the middle of the basal side of integrin mutant FCs contract in an oscillatory fashion. This finding indicates that actomyosin networks do not need to be integrin-linked to the ECM in order to contract. This supports results from cell culture experiments showing localised and stochastic pulses of non-muscle MyoII assembly and disassembly in cells that adhere independently of integrin-ECM engagement [37]. Finally, our results also show that the fibers found in integrin mutant FCs oscillate more stochastically than controls. We speculate that cell-ECM interactions mediated by integrins might also be required to minimise stochasticity, so that the control of cellular tension is guaranteed, thus allowing the harmonised changes in cell shape required for proper epithelia morphogenesis.

Detachment of cells from the substratum by trypsinisation causes a redistribution of F-actin from stress fibers to cortical regions [38]. However, inhibition of cell-matrix adhesion in FCs, by RNAi or optogenetics, was shown to control F-actin intensity, but not its redistribution [18, 19]. In contrast and in agreement with Yamane et al [38], here we show that F-actin decreases in stress fibers but increases in actin rich protrusions in integrin null mutant FCs, suggesting that F-actin could reorganise in the absence of integrins. The difference between our results and those from Qin et al. might reside in the approaches used to eliminate integrin function, knockout in our case (loss of function alleles) versus knockdown in their case (RNAi or optogenetics). The building blocks of stress fibers (actin, myosin and interacting proteins) are constantly exchanged with a cytoplasmic pool. In this context, we propose that the ectopic actin rich protrusions found in integrin mutant cells could arise from an increase in free cytoplasmic actin, due to the reduction in the number of stress fibers, which, upon interaction with membrane actin binding proteins, could organise in protrusions. This view is supported by *in*

*vitro* experiments showing that free actin can organise in different types of networks in bilayers containing membrane-actin linkers [39].

Our results also show that integrins are required for proper growth of the basal surface of FCs, implicating integrin function in cell shape control. Previous studies have proposed that integrins regulate proper egg chamber elongation by acting on actomyosin dynamics [18, 19, 16, 40]. We suggest that integrin function as regulators of cell shape in non-migrating cells could also contribute to the correct shaping of the tissue. Stress fibers are important for the maintenance of shape in crawling cells [41]. However, more controversial is their role for the overall shape of non-migrating cells. Our results show that integrin mutant FCs have reduced number of stress fibers and increased actin rich protrusions and membrane tension. In addition, our findings showing that myosin accumulation correlates with both increased protrusion area and reduced basal surface area in mutant FCs, lead us to propose that the symmetrical and periodic basal surface contractions observed in integrin mutant cells could arise from interactions between the actin-rich protrusions and the medial basal actomyosin. Furthermore, we find that blocking the formation of protrusions in integrin mutant FCs is sufficient to rescue both the increase in membrane tension and the reduction in basal surface. Altogether, our results strongly suggest that the defects in basal surface expansion observed in integrin mutant FCs most likely result from the increase in membrane tension, produced by the ectopic actin protrusions, rather than a consequence of defects in stress fibers. Thus, we propose that integrin function on the assembly and maintenance of stress fibers is crucial to prevent transitions of actin into other types of networks, reorganisations that interfere with the cell shape changes ensuring epithelia development.

We find that clones of integrin mutant FCs undergo further basal surface reduction as oogenesis progresses. Furthermore, this is often associated with the stretching of surrounding control cells and the polarisation of their actin cytoskeleton. Data from cell culture experiments have shown that availability of free space is sufficient to trigger cell migration in the absence of mechanical injury [42]. Here, our *in vivo* analysis shows that control FCs surrounding integrin mutant cells spread their basal surface anisotropically over that of the mutant cells. Furthermore, similar to what happens during asymmetric cell spreading in culture [33], control FCs re-orient their stress fibers towards the integrin mutant cells. Thus, we propose that the activation of the spreading capacity of wild type cells surrounding mutant cells, probably as a response to available free ECM space, could contribute to the additional reduction in cell surface observed in late integrin mutant cells. Alternatively, as mutant cells undergo symmetrical basal surface constriction, they could pull on adjacent wild type cells leading to their stretching and stress fiber reorientation. In this context, the reduction of the basal surface observed in S10B FCs lacking integrins could also arise from the induction of a mechanical response in surrounding wild type cells. Whatever the mechanism, we show here that elimination of integrin function in a group of FCs causes basal constriction and their confinement into the interior of the follicular epithelium by both cell autonomous and non-cell autonomous effects. Of interest for our findings, loss of α2β1 integrin expression results in increased extravasation in breast and prostate cancer [43]. The phenomenon described here may represent a mechanism to facilitate the evasion of tumor cells with low levels of integrins from their original tissue.

Intracellular actin networks can organise in diverse patterns that normally localise to precise regions of the cells. Nevertheless, they are rarely independent and often their dynamics influence each other. Here, we propose that the role of integrins in the maintenance of a specific type of network, stress fibers, is crucial to avoid reorganisation of other actomyosin networks, which we show can lead to defects in cell shape. A wide range of diseases, including cancer and neurological and musculoskeletal disorders, result from uncontrolled actomyosin

networks transitions [44]. Thus, identifying new regulators to restrain transitions between different types of actin networks is crucial to fully comprehend not only morphogenesis but also the cellular and molecular basis of some pathologies.

## Materials and methods

### *Drosophila* stocks and genetics

The following fly stocks were used: *mys¹¹* (also known as *mys^XG43* [45], Sqh-GFP [46], Sqh-mCherry [20, 46] from Bloomington *Drosophila* Stock Centre, UAS-*abi*RNAi (DGRC-Kyoto 9749R), the follicle stem cell driver *traffic jam*-Gal4 (*tj-gal4*, [47]), UAS-zip^DN (a gift from D. Kiehart) and the cell membrane marker Resille-GFP [48]. The *e22c-gal4* driver is expressed in the follicle stem cells in the germarium and was therefore combined with *UAS-flp* to generate *mys* FC clones. To visualise cell membranes in *mys* mutant clones, *mysXG43FRT101/FMZ;* Resille-GFP females were crossed to *ubiquitin*-GFPFRT101; *e22c-gal4 UAS-flp/CyO* males. To analyse myosin dynamics in *mys* mutant clones, *mysXG43FRT19A/FMZ;* Sqh-GFP or *mysXG43FRT101/FMZ;* ResilleGFP:Sqh-mCherry/CyO females were crossed to *nlsRFP FRT19A; e22c-gal4 UAS-flp/CyO* and *ubiquitin*-GFPFRT101; *e22c-gal4 UAS-flp/CyO* males, respectively. To study F-actin distribution and dynamics an ubiquitin- lifeactinYFP construct (described below) was generated and recombined with Resille-GFP. To analyse actin dynamics in *mys* mutant clones, *mysXG43FRT101/FMZ; ubiquitin*-lifeactinYFP:Resille-GFP/CyO females were crossed to *ubiquitin*-GFPFRT101; *e22c-gal4 UAS-flp/CyO* males. To analyse the behaviour of GFP control clones, we crossed FRT101/FMZ; Resille-GFP females to *ubiquitin*-GFPFRT101; *e22c-gal4 UAS-flp/CyO* males. To express *zip^DN* in groups of FCs, we used the mosaic expression system called FLP-OUT [49]. For the rescue experiments, we used the heat shock flipase (*hs-flp*) system [50] to generate follicle cell mutant clones and *tj*-Gal4 to express either UAS-*abi*RNAi or UAS- *zip^DN*/TM2. Females *mysXG43FRT101/FMZ;* UAS-*abi*RNAi/ CyO or *mysXG43FRT19A/FMZ;* UAS- *zip^DN*/TM2 females were crossed to *hs-flpGFPFRT101; tj*-Gal4:Resille-GFP or *hs-flpRFPFRT19A; tj*-Gal4 males. The heat shock was performed at 37ºC for 2 h during third instar larvae and newly hatched females. Flies were kept at 25˚C and yeasted for 2 days prior to ovary dissection.

### *Ubiquitin*-lifeactinYFP construct

Sequences for the actin-binding peptide lifeactin tagged with the fluorescent protein YFP with optimised codon use for *Drosophila* and KpnI and NotI ends were designed. They were synthesised "*in vitro*" by Sigma. The sequences were cloned into the polylinker of the pWR-pUbq transformation vector. This vector contains a poliubiquitin promoter and a selectable marker *mini-white+*. The plasmid was introduced into the germ line of $w^{1118}$ flies by standard methods by the company BestGene Inc. and several independent transgenic lines were isolated.

### Immunohistochemistry

Flies were grown at 25˚C and yeasted for 2 days before dissection. Ovaries were dissected from adult females at room temperature in Schneider's medium (Sigma Aldrich). The muscle sheath that surround ovarioles was removed at this moment. After that, fixation was performed incubating egg chambers for 20 min with 4% paraformaldehyde in PBS (ChemCruz). Samples were permeabilized using PBT (phosphate-buffered saline+1% TritonX100). For actin labelling, fixed ovaries were incubated with Rhodamine Phalloidin (Molecular Probes, 1:40) for 40 min. The following primary antibodies were used: chicken anti-GFP (1/500, Abcam), anti-cDcp1

(1/100, Cell Signaling Technology) and mouse anti-Dlg (1/50, DHSB, Iowa). Fluorescence-conjugated antibodies used were Alexa Fluor 488 and Alex Fluor 647 (Life Technologies). Samples were mounted in Vectashield (Vector Laboratories) and imaged on a Leica SP5 MP-AOBS.

## Time-lapse image acquisition

For live imaging 1–2 days old females were fattened on yeast for 48–96 hours before dissection. Culture conditions and time-lapse microscopy were performed as described in [51]. Ovarioles were isolated from ovaries dissected in supplemented Schneider medium (GIBCO-BRL). Movies were acquired on a Leica SP5 MP-AOBS confocal microscope equipped with a 40 × 1, 3 PL APO oil and HCX PL APO lambda blue 63x 1.4 oil objectives and Leica hybrid detectors (standard mode). $Z$-stacks with 20–23 slices (0.42 μm interval) were taken to capture the entire basal surface of the cells with time points every 30 seconds up to 1 hour.

## Laser ablation

Laser ablation experiments were performed in an inverted Axiovert 200 M, Zeiss microscope equipped with a water-immersion lens (C-Apochromat 633 NA 1.2, Zeiss), a high-speed spinning-disc confocal system (CSU10, Yokogawa), a cooled B/W CCD digital camera (ORCA-ER, Hamamatsu) and a 355 nm pulsed, third-harmonic, solid-state UV laser (PowerChip, JDS Uniphase) with a pulse energy of 20 mJ and 400msec pulse duration. A Melles Griot ArIon Laser (l = 488 nm, 100 mW) was used for excitation of enhanced green fluorescent protein. To analyze the vertex displacements of ablated cell bonds, we first averaged the vertex distance increase from different ablation experiments (ΔL) using as $L_0$ the average of distance of the vertex ten seconds before ablation. Changes in vertex distance before ablation are caused by the movement of cell membranes. Images were taken before and after laser pulse every 0,8 seconds for a period of 10 seconds. The initial velocity was estimated as the velocity at the first time point (t1 = 0,8 s). Standard errors were determined.

## Image processing and data analysis

For quantification of basal myosin and actin dynamics over time, maximal projections of confocal stacks were produced to cover for egg chamber curvature. Integrated intensity of myosin and actin were quantified for manually selected regions using ImageJ software. The background value taken from cell-free regions was subtracted from all data series. Data were subjected to Gaussian smoothening with s = 3, σ = 3. The distribution of oscillation periods was obtained by measuring the intervals between each pair of two adjacent peaks. Actin and myosin intensity changes in both wild type and *mys* mutant cells were obtained by averaging the difference between the maximum and the minimum fluorescence intensity for each oscillation. To calculate the period of myosin and actin oscillation, a Matlab script described in [52] was used to measure the power spectrum density of the signal using one-dimensional Fourier transform of the autocorrelation function.

Number of stress fibers was calculated using ImageJ software. First, the entire basal surface of the cell, excluding the cortical region, was outlined. Then, a line extending across this basal surface, on its central region, perpendicular to the actin bundles was drawn. "Plot profile" tool was employed to quantify the fluorescence intensity of the peaks along this line. To quantify peaks in mutant FCs, only peaks greater than one value of SD below the mean intensity of those found in wild type cells were considered. Number of peaks within a cell divided by the cell area was used to compute peak density (Number of picks/μm). Measurements of whole fluorescence intensity were done by dividing the mean of all included pixels intensity by the

outlined cell area. Since we needed to adjust laser intensity in each sample to properly visualise actin bundles, due to staining heterogeneity, the ratio between the fluorescent intensities of *mys* and wild type cells was plotted.

Cortical actin intensity across cell-cell boundaries was measured by quantifying the intensity of fluorescent signal across a 2μm bar centered at the boundary. The total fluorescence intensity of the bar was normalised with respect to wild type values.

Measurement of total protrusion area was done manually as described previously in [53, 54]. In brief, protrusion area was calculated in FCs expressing the membrane marker Resille-GFP by drawing freehand Regions of Interest (ROIs) from the cell surface to the tip of the protrusions, as shown in orange in Fig 2D, and calculating total protrusion area as a percentage of total basal surface area.

Cell area data were calculated using Imaris (Bitplane). The whole basal surface of the cell was outlined using Resille-GFP as cell membrane marker.

## Supporting information

**S1 Movie. Dynamics of whip-like structures in S7 control and *mys* FCs.** Time-lapse movie of a rotating S7 mosaic egg chamber containing *mys* FC clones and expressing LifeactinYFP (Ubi-LifeactinYFP, red) and Resille-GFP (green). Focus is on the basal surface. *mys* FCs (GFP-negative, yellow arrows) contain a higher number of whip-like structures compared to controls (GFP-positive, white arrows), which behave similar to controls, i.e. propelling against the direction of rotation.
(MOV)

**S2 Movie. Dynamics of myosin in S8 control and *mys* FCs.** Time-lapse movie of a S8 mosaic egg chamber containing mys FC clones and expressing Sqh-GFP Focus is on the basal surface Myosin does not decorate whip-like structures in either control (nuclear RFP-positive) or mys (nRFP-negative) FCs Dots correspond to aggregates of the Sqh-GFP protein.
(MOV)

**S3 Movie. Basal F-actin oscillations in live S10 *mys* FCs.** Time-lapse movie of a S10 mosaic egg chamber containing *mys* FC clones and expressing LifeactinYFP (Ubi>LifeactinYFP, red) and Resille-GFP (green). Focus is on the basal stress fibers. Note that F-actin can oscillate in *mys* FCs (GFP-negative).
(MOV)

**S4 Movie. Basal myosin oscillations in live S10 *mys* FCs.** Time-lapse movie of a S10 mosaic egg chambers containing *mys* FC clones and expressing Sqh-mCherry (red) and Resille-GFP (green). Focus is on the basal stress fibers. Note that myosin can oscillate in *mys* FCs (GFP-negative).
(MOV)

**S5 Movie. Integrin mutant cells show abnormal dynamic actin protrusions.** Time-lapse movie of a S10 mosaic egg chamber containing *mys* FC clones and expressing LifeactinYFP (Ubi-LifeactinYFP) and Resille-GFP (green). Focus is on the basal surface. Note the presence of dynamic F-actin protrusions (white arrow), emerging from the cell cortex and projecting towards the cell center in *mys* FCs (GFP-negative).
(MOV)

**S6 Movie. Dynamics of actin protrusions, myosin levels and basal surface contractions in *mys* FCs.** Time-lapse movie of a S10 mosaic egg chambers containing *mys* FC clones and expressing Sqh-mCherry (red) and Resille-GFP (green). Focus is on the basal surface. Note the

correlation between increased protrusion area, myosin accumulation and decreased basal surface area in *mys* FCs.
(MOV)

**S7 Movie. Laser ablation of cell bonds between wild type cells.** Movie corresponds to the ablation experiment shown in Fig 3. The membranes of FCs are visualised with Resille-GFP. A cell bond between two control FCs is ablated. GFP fluorescent is lost in the middle of the ablated bond upon laser ablation. The movie continues 15s after the cut and shows displacement of the vertexes. Images are taken every 0.8 seconds.
(MOV)

**S8 Movie. Laser ablation of cell bonds between *mys* mutant cells.** Movie corresponds to the ablation experiment shown in Fig 3. The membranes of FCs are visualised with Resille-GFP. A cell bond between two *mys* FCs is ablated. GFP fluorescent is lost in the middle of the ablated bond upon laser ablation. Movie length and frame rate are as described for S7 Movie.
(MOV)

**S9 Movie. Dynamic behavior of the basal surface of control FCs contacting mutant FCs.** Time-lapse movie of a S10 mosaic egg chambers containing *mys* FC clones and expressing Resille-GFP (green). Focus is on the basal surface. Note that the basal surface of control FCs (GFP-positive) contacting *mys* FCs (GFP-negative) seems to spread over the basal surface of the mutant ones.
(MOV)

**S10 Movie. Dynamic behavior of the basal surface of control FCs.** Time-lapse movie of a S10 mosaic egg chambers containing GFP FC clones and expressing Resille-GFP (green). Focus is on the basal surface. Note that the basal surface of control FCs (GFP-positive) enclosing GFP mutant FCs (GFP-negative) does not spread over the GFP mutant FCs.
(MOV)

**S1 Fig. *mys* FCs do not die by apoptosis. (A)** Basal surface view of a mosaic S10 egg chamber containing *mys* FC clones stained with anti-GFP (green), anti-Dcp-1 (red) and the nuclear marker Hoechst (blue). Scale bar, 20 μm. **(A', A")** Magnifications of the white box in **A**. Scale bars, 10 μm.
(TIF)

**S2 Fig. Control and *mys* whip-like structures do not contain myosin. (A, B)** Confocal images, taken with a 30 min. interval, of a live rotating S8 mosaic egg chamber containing *mys* FC clones (nuclear RFP-negative) and expressing Sqh-GFP (green). Arrow in A indicates the direction of egg chamber rotation. **(A', A", B' and B")** Magnifications of the white boxes in **A** and **B**, respectively. Asterisks label a cell as a reference for the rotation. Dots correspond to aggregates of the Sqh-GFP protein (yellow arrow in A"). Scale bars, 20μm in **A** and **B** and 5μm in **A', A", B'** and **B"**.
(TIF)

**S3 Fig. Stress fibers in *mys* FCs show reduced myosin compared to controls. (A, B, C)** Basal surface view of mosaic S8 **(A, A')**, S9 **(B, B')** and S10 **(C, C')** egg chambers containing *mys* FC clones, expressing Sqh-GFP (green) and stained for anti-RFP (red). **(A-C')** Myosin levels in stress fibers diminish progressively from S8-10 in *mys* FCs (RFP-negative). White and yellow arrows point to stress fibers in control (RFP-positive) and mutant FCs, respectively. **(D)** Quantification of relative myosin levels in stress fibers in control and *mys* FCs. Scale bars, 5μm.
(TIF)

**S4 Fig. Loss of integrins affects the levels and dynamics of basal F-actin. (A, B')** Confocal images, taking with a 15 min interval, of live S10 egg chambers containing *mys* FC clones and expressing LifeactinYFP (red) and the cell membrane marker Resille-GFP (green). White and yellow arrows point to stress fibers in control (GFP-positive) and mutant (GFP-negative) FCs, respectively. **(C)** Quantification of the dynamic changes of basal F-actin intensity in S10 control (green) and *mys* FCs (grey). **(D, E)** Fourier transform of the autocorrelation function of the temporal sequences of basal F-actin intensity for control **(D)** and *mys* **(E)** FCs. T indicates period of oscillations. Scale bars, 5 μm.
(TIF)

**S5 Fig. Elimination of integrins affects the levels and dynamics of basal myosin. (A, B')** Confocal images, taking with a 15 min. interval, of live S10 egg chambers containing *mys* FC clones and expressing Sqh-mCherry (red) and the cell membrane marker Resille-GFP (green). **(A', B')** White and yellow arrows point to stress fibers within control (GFP-positive) and mutant (GFP-negative) FCs, respectively. **(C)** Quantification of the dynamic changes of basal myosin intensity in S10 control (green) and *mys* (grey) FCs. **(D, E)** Fourier transform of the autocorrelation function of temporal sequences of basal myosin intensity for control **(D)** and *mys* FCs **(E)**. T indicates period of oscillations. Scale bars, 5 μm.
(TIF)

**S6 Fig. Cortical F-actin increases autonomously and specifically on the basal surface of *mys* FCs. (A-B')** Basal **(A, A')** and apical **(B, B')** surface views of S9 mosaic egg chambers containing *mys* FC clones, stained for anti-GFP (green) and Rhodamine Phalloidin to detect F-actin (red). Basal **(A, A')**, but not apical **(B, B'**), cortical actin levels are higher in *mys* FCs (GFP-negative, yellow arrow) compared to controls (GFP-positive, white arrow). **(C)** Quantification of relative cortical actin intensity in control and *mys* FCs. **(D)** Histogram of fluorescent intensities of F-actin along boundaries between control and *mys* FCs, as indicated with straight coloured lines in **(A)**. Scale bars, 5 μm.
(TIF)

**S7 Fig. Expression of a dominant negative form of zip (zip$^{DN}$-GFP) rescues the reduced basal surface found in *mys* FCs. (A, A')** A S10 mosaic egg chamber containing clones of FCs expressing zip$^{DN}$-GFP and stained with anti-GFP (green) and Rhodamine Phalloidin to detect F-actin (red). **(B, B')** Mosaic egg chamber expressing zip$^{DN}$-GFP in all FCs and containing *mys* FC clones (GFP-negative) stained with anti-GFP (green), Rhodamine Phalloidin (red) and anti-Dlg (blue).
(TIF)

**S1 Data. Numerical data underlying graphs in Fig 1. (A)** Whip-like structures (Fig 1F). **(B)** Number of peaks/μm (Fig 1G). **(C)** Relative actin intensity in stress fibres (Fig 1H).
(TIF)

**S2 Data. Numerical data underlying graphs in Fig 2.** Basal area occupied by projections, basal surface area, myosin intensity (Fig 2E).
(TIFF)

**S3 Data. Numerical data underlying graphs in Fig 3. (A)** Initial velocity vertex displacement (Fig 3C). **(B)** Vertex distance increase (Fig 3D).
(TIF)

**S4 Data. Numerical data underlying graphs in Fig 4.** (**A**) Apical area (Fig 4E). (**B**) Height (Fig 4F). (**C**) Basal area through development (Fig 4G). (**D**) Nuclear size (Fig 4H).
(TIF)

**S5 Data. Numerical data underlying graphs in Fig 5.** (**A**) Cortical actin fluorescence intensity ratio (Fig 5C). (**B**) Basal area rescue (Fig 5D). (**C**) Initial Velocity vertex displacement rescue (Fig 5E).
(TIF)

**S6 Data. Numerical data underlying graphs in S3D Fig.** Relative myosin intensity (S3D Fig).
(TIF)

**S7 Data. Numerical data underlying graphs in S4C Fig.** Basal Actin Oscillation (S4C Fig).
(TIF)

**S8 Data. Numerical data underlying graphs in S5C Fig.** Basal myosin oscillation (S5C Fig).
(TIF)

**S9 Data. Numerical data underlying graphs in S6C and S6D Fig.** Cortical actin fluorescence (S6C and S6D Fig).
(TIF)

## Acknowledgments

We thank the Bloomington and Kyoto Stock Centres for fly stocks and reagents. We are grateful to A. Gonzalez-Reyes for useful comments on the manuscript.

## Author Contributions

**Conceptualization:** María D. Martín-Bermudo.

**Formal analysis:** Carmen Santa-Cruz Mateos, Andrea Valencia-Expósito, María D. Martín-Bermudo.

**Funding acquisition:** María D. Martín-Bermudo.

**Investigation:** Carmen Santa-Cruz Mateos, Andrea Valencia-Expósito, María D. Martín-Bermudo.

**Methodology:** Carmen Santa-Cruz Mateos, Andrea Valencia-Expósito, María D. Martín-Bermudo.

**Project administration:** María D. Martín-Bermudo.

**Resources:** María D. Martín-Bermudo.

**Supervision:** María D. Martín-Bermudo.

**Validation:** María D. Martín-Bermudo.

**Writing – original draft:** María D. Martín-Bermudo.

**Writing – review & editing:** Isabel M. Palacios, María D. Martín-Bermudo.

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
