## [Decision Letter · Decision Letter 0]

4 Oct 2019

Dear Lola,

Thank you very much for submitting your Research Article entitled “Integrin-mediated cell-ECM interactions regulate epithelia morphogenesis and homeostasis by controlling the architecture and mechanical properties of basal actomyosin networks” to PLOS Genetics. Your manuscript was fully evaluated at the editorial level and by three independent peer reviewers. The reviewers appreciated the attention to an important problem, but raised very important concerns about the current manuscript. Based on the reviews, we will not be able to accept the manuscript, at least in its current version. However, we would be willing to review again a much-revised version including experimental work. We cannot, of course, promise publication at that time. The most important concerns are:

Novelty. Please trim down your manuscript to novel results only. Even nice but confirmatory observations should be eliminated or put in supplementary files.Title. Several reviewers point out that the manuscript does not address “epithelial homeostasis” or “morphogenesis”. I suggest to remove "tissue scale" conclusions from the title and elsewhere in the manuscript.Cortical actin: reviewer1 wonders whether the increase in cortical actin is a secondary consequence of the lack of stress fibers in integrin mutant clones, and should be presented as such.

Reviewer3 is not convinced by the nature of cortical actin, and would like a better description.

Several reviewers are not convinced by the comparison with wound healing.

We feel that these experiments should be performed before re-submission.

Should you decide to revise the manuscript for further consideration here, your revisions should address all the specific points made by each reviewer. We will also require a detailed list of your responses to the review comments and a description of the changes you have made in the manuscript.

If you decide to revise the manuscript for further consideration at PLOS Genetics, we would appreciate an expected resubmission date by email to plosgenetics@plos.org.

[LINK]

We are sorry that we cannot be more positive about your manuscript at this stage. Please do not hesitate to contact us if you have any concerns or questions.

Yours sincerely,

Jean-René Huynh

Associate Editor

PLOS Genetics

Gregory P. Copenhaver

Editor-in-Chief

PLOS Genetics

Reviewer's Responses to Questions

**Comments to the Authors:**

Reviewer #1: The work of Santa-Cruz Mateos et al explores the role of integrins in Drosophila follicle cells. These cells are a great example of stress fibers that can be studied in vivo, with all the power of genetics and imaging brought by this model. Authors used mosaic tissues for a mutation abolishing integrin complex and analyzed its impact on the actomyosin both on fixed and live samples. Although the authors made some nice observations, as it stands, I do not consider that this manuscript reaches the criteria for publication in PLos Genetics. I have several concerns regarding the data or their interpretation.

1) The article suffers of a lack of novelty. Most of the observations confirm existing data, including on the same tissue, both on F-actin cytoskeleton aspect and actomyosin oscillatory behavior. The best illustration of this is given by the title itself, extremely long but with no clear message: that integrins are important in epithelial cells and have something to do with actomyosin does not seem really new. Title also mentions epithelium morphogenesis and homeostasis, but I cannot see where these points are really addressed in this work. Actually, authors have never looked at a tissue scale.

2) I am also a bit puzzled by some interpretations :

The main novelty would be that integrin modulates other actin populations as cortical actin and “whips” (figs 1 and 3) . Without integrin, cells produce much less stress fibers (fig 2) and more G-actin is available for cortical F-actin and whips, probably explaining why more of these populations is observed. This raise in cortical actin increases cell tension and induces basal constriction. All data point to this interpretation and authors also reach this conclusion in their discussion and their abstract. Thus, it seems a secondary effect of integrin loss and it is therefore difficult to claim, as authors did several times, that integrin “modulates” or “controls” cortical actin. It would be actually more logical to present the effect of integrin on stress fibers (fig 2) in first. Moreover, whip-like structures are so far poorly characterized at the functional level and depends on the same actin regulators than cortical F-actin; these data could be fused with the ones of fig3. Overall, the figures could be simplified as they show many redundancies. For instance, there are five pictures of F-actin in mys clones around stage 8-9 on four different figures. One could advice the authors to change their manuscript in a shorter report with four figures, highlighting what is really new.

Also, authors stated that wild-type cells next to the mutant ones “acquire spreading capacity” and compared this property with wound healing. It seems highly speculative and even maybe contradictory with their results. Authors have shown that cortical tension increases in mutant cells and that it is associated with basal surface reduction. It is likely that this basal constriction pull on the adjacent cells because of cell-cell adhesion and that stress fibers point towards the mutant cells as a consequence of this mechanical stress. If the wild-type cells were actively spreading over the mutant ones, it would somehow press them, which should decrease cortical tension in the mutant cells. Thus, it seems clearly different than a wound healing process.

These two points are really important because the main integrin primary function described so far in these cells is to allow basal surface contraction, at least transitorily. Thus, if not explained properly it might introduce confusion in the field.

3) An other important drawback is about the quantitative data throughout the paper.

- Some are missing. Fig 1: whip-like structures are not quantified. Fig 6 : I really like these observations but there is no indication of cell frequency losing their basal domain and how frequent the reorientation of the surrounding is. It is also true for several supplemental figures ( S1,S2 and S8 for instance).

- For some of them, no statistic test is provided : 5D, 5E, 4G

- For some their description is incomplete or approximative and does not allow a clear interpretation. Fig 2D and E: How authors can discriminate long protusions of cortical actin versus stress fibers, especially in mutant cells where less (no?) stress fibers are present and in which protusions are more preeminent due to the increase of cortical actin (visible on fig 1D for instance). Fig 3 : According to the graphs, laser cut data look very robust. However, no pictures or movies are shown. Moreover, the genotype used for these experiments is not indicated. Also, it is not described how the choice of the cut is made in a mutant clone. I guess the bound is between two mutant cells but what about the two neighboring cells at bound extremities: are they wildtype? are they mutant ? are the results the same in both cases ?

- For some quantitative data, the results seem not consistent with other author observations : How authors can measure basal surface with such a small error bar at stage 10 (fig 4G) whereas this value is highly variable during this stage and can even reach zero for some cells according to fig 6.

4) Finally, some experiments could be added to fulfill their study. Authors nicely show that Abi RNAi rescues cortical actin levels and basal surface. Laser cuts on such phenotype could fulfill the demonstration that cell surface diminution is due to the increase of cortical actin tension. Moreover, Do they observe wild-type adjacent cells reorienting their stress fibers around mys, Abi RNAi cells. If not that will tend to confirm that it is a response to the mechanical stress induced by mys mutant cells.

Minor points :

Delon and Brown nicely described some changes in integrin complex composition depending on the stage. Could these differences explain why the mutant cells lose their contact specifically at stage 10?

Introduction : “In summary, F-actin organizes in three different types of networks at the basal side of FCs. “ This sentence is difficult to understand as the paragraph mainly describes stress fibers and that the two others populations are mentioned briefly and much earlier. Actually, a scheme explaining the different actin populations could help.

The only argument of the authors may have for a direct impact of integrin on cortical actin is on fig1C where a localization at, or close to, tricellular junctions is observed. First, this localization of integrin at tricellular junction has already been shown in Schotman et al 2008, in which authors demonstrated that this localization is more a step of their trafficking than a functional site. Then, if such localization will have something to do with the cortical actin defect observed in mys clones, then it should be observed before stage 10. Is it the case?

Some pictures do not seem oriented according to AP axis, as it is usually the case (fig 2 for instance). This is a bit disturbing as stress fibers are normally planar polarized in this tissue.

Authors stated that MyoII signal is not increased in mutant cells fig S1 and movie S2. Despite the fact that there is no quantification, I do even see less, probably due to the absence of stress fibers where MyoII localized during the rotation process. To this respect Viktorinova et al, 2017 should be cited.

Fig 2C: F-actin looks extremely weird in WT cells. The fact that they have been generated by STED cannot explain such a difference with usual confocal images.

Authors tried to detect apoptosis with cleaved cas3 staining. It is now well established that dcp1 staining is a more reliable marker in fly.

In the pictures there is a mix of different stages that make very difficult for a non-specialist; Why authors look at stage 8, 9 or 10 (but not the others)?

Lack of figure annotation makes sometimes article difficult to read :

Fig 2E and 3E: the two graphs represent in appearance the same thing with opposite results.

- Fig 3C, 4C : what is plotted on the x-axis?

- S3G : what is plotted on the x-axis? What is the difference between the two graphs ? is it time ? if so I do not understand why MyoII and actin would be anticorrelated (whereas they were correlated in the precedent simulations published by the same group.)

Beginning of page 12 : “(or Fat)”?

Discussion : Qin paper that showed by optogenetics that integrin are not required for the oscillatory behavior of the cells should be mentioned in the discussion. Actually, I would recommend the authors to show their data on MyoII pulsation in the main figures. Although it comes as confirmation, I find amazing to see that, even in null mutant cells, such oscillation still occurs.

Reviewer #2: Summary:

This study characterizes the effects of the mysXG43 mutation on the FC cytoskeletal elements – cortical actomyosin fibers, stress fibers, and the whip-like structures at the TCJs, expanding on the knowledge of integrins and their functions in the FE. Previously shown to have increased F-action levels in general, the integrin deficient mutant mysXG43 is now shown to increase F-actin specifically at the whip-like protrusions at the TCJs and the cortical actomyosin network, but NOT at the stress fibers. Integrins are reported to be involved in the setting up, functioning, and maintenance of the cortical actomyosin network as the egg chambers develop from stage 8 – 10. By laser ablation of cell-cell contacts in mutant mosaic FE, they show that the cell membranes are at a higher tension in the mutant cells than in the WT cells. Finally, they show that integrins work with Abi to expand the basal surface area of the endocycling FCs. In the mosaics, they also report a rearrangement of the mutant cells with reduced (or almost absent/sunken) basal membrane, between the WT cells in the FE. This study increases our understanding of the role of integrins in maintaining and regulating the FE actomyosin dynamics.

Comments:

General comments-

- The result section has a lot of ‘introduction’ at the beginning of each sub-topic, and can be moved to the intro section of the paper to make the data easier to follow.

- Discussion section has a lot of grammatical errors. Proofread to improve. Good section, though.

- The title mentions ‘epithelia morphogenesis and homoeostasis’ but I don’t really see any evidence for this in the data. There is no real evidence for this claim – yes, the mutant cells have an altered morphology (smaller basal surface) but it is the WT cells that compensate for this, and integrins are not shown to have a role in it. And yes, integrins regulate the cytoskeleton and therefore cell shape, but not epithelial morphogenesis per se. Consider rewording so as to not appear to be misleading.

- Figure legends also need proofreading to make them more clear – especially the supplementals.

- Egg chamber stages for ALL figures need to be specified – even in the supplementary figures.

- The movies are awesome.

Specific comments –

- Page 4 – Introduction, paragraph 3 – ovarioles are described as ‘tube-producing eggs’. Please rephrase.

- Figure 1 A-A” – What is the purpose of showing the localization of integrins at stage 10a? Make a note of why this is important either here or in the discussion.

- Is the F-actin stained by phalloidin in stage 8 egg chambers in Fig 1D-D” basal cortical actin, and stress fibers in Fig 2 A-A’? There is a need to distinguish between the two and how you tell them apart, at least in the introduction. Otherwise it gives the impression of conflicting data between your two figures.

- Page 9 – under “Integrins regulate cortical F actin levels and tension” – the first sentence either needs a reference or can be removed altogether. You can go straight to describing your experiment/data.

- Page 10 – under “Morphological consequences…” – The first paragraph contains an excessive amount of introduction or background that is hard to read and, again, distracts from the actual data. Condense for better flow, and for a clearer storyline.

- Page 11-12 – under “Integrins control basal surface…” zipperDN is reported to have reduced stress fiber numbers. Please include a note on what this signifies in this section, and how it fits in your hypothesis, as you did with the Abi RNAi data.

- Page 13 – Discussion – paragraph 1 line 6-7 – “…this integrin function is also critical for the changes in cell shape and area underlying epithelia morphogenesis.” Epithelial morphogenesis indicates a different process in drosophila egg chambers at different stages. Which of those is affected in the mysXG43 mutant cells? Or are you referring generally to cell shape changes caused by the mutation to be ‘epithelial morphogenesis’? Please clarify, as this comes off as misleading to some extent.

- Page 27 – figure legend –figure 5 – “integrins regulate cell surface growth…” Your data indicates that integrins are required for the normal increase in basal surface area – your figure legend is misleading.

- Page 27 – figure legend – figure 6 – “Elimination of integrins in FCs induces spreading of the basal surface of control neighboring FCs” This description indicates that it is the mutation that is somehow ‘inducing’ the spreading of the WT cells in the mosaic. I would argue that it is the phenotype created by the mutation that is eliciting the response from the WT cells in the mosaic (potentially to maintain tissue integrity). Please correct this description.

- Page 28 – Figure legend – Movie S2 – “The whip-like structures found in mysXG43 FCs do not contain myosin” is the title, and you go on to say two lines down that it is the same for both control and mysXG43 mutant cells. It is rather misleading to word the title this way – it makes more sense to say that the mutation does not affect myosin distribution in the whip-like structures. Please correct.

- Page 29/Movie S5 – The movie is great! However, it would be easier to see the basal domain of the WT cells expanding if there was an arrow or arrowhead to point out where to look. (The control in Movie S6 helps).

Also, there is a cell doing a lot of moving, almost like a junction remodeling event on the top left corner – please address that with a hypothesis or speculation.

- Page 30 - figure legend – supplementary figure 5 – “…specific of the basal surface” do you mean restricted to the basal fibers?

Reviewer #3: Uploaded as attachment.

**Have all data underlying the figures and results presented in the manuscript been provided?**

Reviewer #1: Yes

Reviewer #2: None

Reviewer #3: Yes

PLOS authors have the option to publish the peer review history of their article (what does this mean?). If published, this will include your full peer review and any attached files.

Reviewer #1: No

Reviewer #2: No

Reviewer #3: No

---

## [Decision Letter · Decision Letter 1]

13 Feb 2020

Dear Lola,

Your revised manuscript has been evaluated by the same three reviewers. Reviewers 2 and 3 are happy with your additional experiments and changes. However, both of them noticed many typos and grammatical errors. It is important that you correct those before I can formally accept the manuscript. 

Reviewer 1 remains critical. I won't send back your manuscript for review, however, I very much advise you to take into consideration its remaining concerns. It won't be a condition for acceptance, but it is an opportunity to improve the published version of your manuscript. Please list any changes, that you decide to make in a separate letter.

[LINK]

Yours sincerely,

Jean-René Huynh

Associate Editor

PLOS Genetics

Gregory Copenhaver

Editor-in-Chief

PLOS Genetics

Reviewer's Responses to Questions

**Comments to the Authors:**

Reviewer #1: The manuscript is improved but I am still not fully convinced about the novelty and the relevance of this work. I guess the question is whether the characterization of secondary effects of integrin loss can be considered as “new roles for integrins” as authors claim in their rebuttal.

1) I still do not find the title in line with the paper as it is presented : the title mentions actomyosin whereas myosin is not looked at anymore in the main figures, which I think is unfortunate.

2) Authors wrote:

“In addition, we noticed that the basal stress fibers, which are randomly oriented in wild type S10B FCs (yellow asterisks in Fig.6E, E’, Delon and Brown, 2009), polarized towards the mutant cells in control FCs surrounding mys FCs (n=15, ec=8, yellow arrows in Fig.6E, E’). Quantification of this phenotype revealed that control cells re-orient their stress fibers around clones of mutant cells in 80% of the cases analysed (n=15, ec=9). “

As I understand it, the same quantification is given twice but not consistently …

Moreover few lines before authors indicate that “ Our analysis of later egg chambers (stage 10b) showed that 70% of the mutant clones (n=30, ec=8) contained cells in which the basal area was hardly visible (Fig.6A-B’) “

But then why number are different (30 in one case 15 in the other) and to what the 80% refer: is 80% of the 70% or 80% of the totality? It may change quite a lot the interpretation.

3) laser cuts movies are nice but on the provided examples I cannot see the difference in the recoil between the two genotypes. Also the duration of the movies and the image size (different scales ?) are different, making theme difficult to compare.

4) Fig 5E : values of recall velocities for WT/WT cells and mys/mys cells look identical to the ones of Fig 3. If the authors reused the same data, it should be indicated. Moreover the n value is not indicated for mys, abi RNAi cuts.

5) Whether Abi RNAi could rescue tension increase and stress fiber orientation in neighboring cells should have tested using MARCM approach. 1) it would have clearly shown that the effect on tension is cell autonomous 2)The general effect of Abi RNAi on stress fiber orientation because it blocks rotation could have been anticipated (already published) to choose a better strategy.

6) Image quality of the PDF is poor, making very difficult to look at some figures.

Reviewer #2: The responses to the original comments are acceptable. However, the manuscript still contains many grammatic errors and needs to be proofread thoroughly.

Reviewer #3: review is uploaded as attachment.

**Have all data underlying the figures and results presented in the manuscript been provided?**

Reviewer #1: Yes

Reviewer #2: Yes

Reviewer #3: No: Spreadsheet for numerical data is missing.

PLOS authors have the option to publish the peer review history of their article (what does this mean?). If published, this will include your full peer review and any attached files.

Reviewer #1: No

Reviewer #2: No

Reviewer #3: No

---

## [Editor Report · Decision Letter 2]

16 Mar 2020

Dear Lola,

Thank you for taking the time to address the reviewers' comments. 

We are pleased to inform you that your manuscript entitled "Integrins regulate epithelial cell shape by controlling the architecture and mechanical properties of basal actomyosin networks" has been editorially accepted for publication in PLOS Genetics. Congratulations!

Yours sincerely,

Jean-René Huynh

Associate Editor

PLOS Genetics

Gregory P. Copenhaver

Editor-in-Chief

PLOS Genetics

Comments from the reviewers (if applicable):

**Data Deposition**

http://datadryad.org/submit?journalID=pgenetics&manu=PGENETICS-D-19-01440R2

**Press Queries**

---

## [Editor Report · Acceptance letter]

15 Apr 2020

PGENETICS-D-19-01440R2 

Integrins regulate epithelial cell shape by controlling the architecture and mechanical properties of basal actomyosin networks 

Dear Dr Martin-Bermudo, 

We are pleased to inform you that your manuscript entitled "Integrins regulate epithelial cell shape by controlling the architecture and mechanical properties of basal actomyosin networks" has been formally accepted for publication in PLOS Genetics! Your manuscript is now with our production department and you will be notified of the publication date in due course.

With kind regards,

Jason Norris

PLOS Genetics

On behalf of:
